

# Dysregulated lipid metabolism in late preterm low birth weight neonates: A case-control study on maternal lipid levels and early metabolic programming implications

Jing Liu[1,2,*], Lijuan Tang[1,3,*], Qi Sun[1], Di Lv[1,2], Yuanmei Chen[1], Fang Ye[1], Die Liu[1], Qin Hui[1], Haixiao Liang[1], Chao Wang[1] and Qi Zhang[1,2]

[1] Department of Pediatrics, China-Japan Friendship Hospital, Beijing, China
[2] Graduate School of Peking Union Medical College, Chinese Academy of Medical Sciences, Bejing, China
[3] Institute of Clinical Medical Sciences, China-Japan Friendship Hospital, Beijing, China
[*] These authors contributed equally to this work.

Corresponding author
Qi Zhang, zhangqikeyan@163.com

## ABSTRACT

**Objectives**. While low birth weight (LBW) is a recognized risk factor for adult metabolic syndrome, the unique lipid metabolic phenotype of late preterm low birth weight (LPTB-LBW) neonates—who experience dual exposures to shortened gestation and intrauterine growth restriction—remains uncharacterized. This study specifically examines whether the convergence of prematurity and growth restriction synergistically disrupts lipid metabolic programming.

**Methods**. Using ultra performance liquid chromatography-tandem mass spectrometry (UPLC-MS/MS), we compared lipidomic profiles of 88 plasma samples: 45 LPTB-LBW ($34^{0/7}$-$36^{6/7}$ weeks, <2,500 g) and 43 later preterm birth-normal birth weight (LPTB-NBW, $34^{0/7}$-$36^{6/7}$ weeks, 2,500–4,000 g) controls. Multivariate orthogonal partial least squares-discriminant analysis and univariate modeling identified discriminant lipids. Maternal-neonatal lipid continuity was assessed through Spearman's correlation analysis.

**Result**. A total of 1,173 lipids were identified, categorized into five major lipid classes, with 349 significantly different lipids detected (324 upregulated and 25 downregulated) in the LPTB-LBW group. All glycerolipids were upregulated, accounting for 50% (162/324) of the upregulated lipids. Long-chain polyunsaturated triglycerides (TG) showed extreme elevations, such as TG (18:2_18:3_18:4) and TG (18:2_20:4_20:5). Monoglycerides, including MG (18:2) and MG (18:1), were also significantly elevated. Among glycerophospholipids (GP), 76 species were upregulated, with notable increases in phosphatidylethanolamines such as PE (O-18:0_22:3) and PE (18:2_22:1), while PG (20:4_22:6) was significantly reduced. All differentially expressed ceramides, including Cer (d26:3/33:1(2OH)), Cer (d29:2/30:2(2OH)), and Cer (d28:3/31:1(2OH)), were upregulated, whereas sphingosines were downregulated. Cholesterol esters were decreased, while bile acids, free fatty acids and acylcarnitines were elevated. KEGG pathway enrichment analysis highlighted significant perturbations in cholesterol, glycerolipid, and sphingolipid metabolism. Maternal high-density lipoprotein cholesterol (HDLC) levels during early pregnancy showed exclusive negative correlations with
neonatal lipids, particularly triacylglycerol TG (16:0_18:2_18:2) ($r = -0.33$, $p = 0.002$), diacylglycerols, and ceramides, whereas no associations were observed for maternal low density lipoprotein (LDLC), TC, or TG.

**Conclusions**. LPTB-LBW neonates exhibit a unique lipidomic phenotype marked by hyperaccumulation of glycerolipids (*e.g.*, long-chain polyunsaturated TGs), elevated ceramides, and altered phospholipid species (increased PE, decreased PG). Maternal HDLC levels negatively correlated with specific neonatal lipids. These findings highlight early-life lipid alterations in LPTB-LBW infants and the need for further investigation into their clinical implications.

# INTRODUCTION

Low birth weight (LBW) poses a growing public health challenge in developing countries, with evidence linking it to adult-onset metabolic and cardiovascular conditions (*Wang et al., 2022*). Despite a modest global decline in LBW prevalence (annual reduction rate: 1.2%, 2000–2015), current progress remains insufficient to achieve the World Health Assembly's target of a 30% reduction by 2025 (*Blencowe et al., 2019*; *United Nations Children's Fund, 2023*). Among LBW infants, late preterm birth-low birth weight (LPTB-LBW) neonates-defined as those born between 34 0/7 and 36 6/7 weeks of gestation with a birth weight under 2,500 g_represent a critical subgroup exposed to dual developmental insults: shortened gestation and intrauterine growth restriction (IUGR). IUGR, driven by maternal malnutrition, placental dysfunction, and comorbidities, constrains fetal growth trajectories and elevates lifelong cardiometabolic risks (*Bendix, Miller & Winterhager, 2020*). Nevertheless, the unique metabolic phenotype of LPTB-LBW neonates remains poorly characterized, hindering targeted interventions for this vulnerable population.

The Developmental Origins of Health and Disease (DOHaD) posits that adverse intrauterine environments reprogram organogenesis, elevating lifelong disease risks (*Buklijas & Al-Gailani, 2023*). Epidemiological studies consistently associate LBW with neonatal mortality, childhood stunting, cognitive deficits, and an elevated risk of adult-onset obesity, type 2 diabetes, and cardiovascular disease (*Bianco-Miotto et al., 2017*; *Feng, Osgood & Dyck, 2018*; *Halli, Biradar & Prasad, 2022*). Notably, LPTB neonates face a 2-3-fold increased risk of acute complications (*e.g.*, hypoglycemia, respiratory distress) and healthcare utilization (*Sharma et al., 2021*). Emerging mechanistic studies implicate dysregulated lipid metabolism as a central mediator of these outcomes. Perturbations in lipid storage (*e.g.*, triglycerides), membrane signaling (*e.g.*, phospholipids), and bioactive sphingolipids (*e.g.*, ceramides) have been linked to insulin resistance and inflammatory cascades in developmental programming models (*Chaurasia & Summers, 2021*; *Povel et al., 2011*). Yet, despite their clinical relevance, comprehensive lipidomic phenotyping of LPTB-LBW neonates is conspicuously absent from the literature.

Recent advances in lipidomics have revolutionized our understanding of neonatal metabolic programming. These studies underscore the critical role of lipid metabolism in fetal development, neurogenesis, and immune function (*Delhaes et al., 2018*). Studies reveal that lipid-related pathways (*e.g.*, linoleic and arachidonic acid metabolism) are disrupted in both SGA and LGA infants, with U-shaped metabolites (*e.g.*, cuminaldehyde) and linear metabolites (*Zhai et al., 2023*). Animal models further highlight dysregulated fatty acid metabolism and PPARα/CYP4A14 signaling in LBW individuals (*Li et al., 2018*; *Zhou et al., 2023*), while human cohorts demonstrate persistent metabolic abnormalities (*e.g.*, propionylcarnitine) tied to insulin resistance in adulthood (*Metrustry et al., 2018*). MR-based metabolomic analyses of IUGR pregnancies reveal a disrupted maternal-fetal lipid axis, characterized by maternal hypocholesterolemia and fetal accumulation of atherogenic lipoproteins, suggesting a transgenerational programming effect *Miranda et al. (2018)*. Urinary metabolomics have identified distinct signatures associated with various neonatal outcomes, including IUGR and bronchopulmonary dysplasia (BPD) (*Dessi et al., 2011*; *Fanos et al., 2014*). For example, studies have observed a significant decrease in histidine levels and the ornithine/citrulline ratio in the BPD group. Additionally, the ratios of acylcarnitines C3/C0 and C5/C0 were also significantly reduced (*Guo et al., 2024b*). While these studies provide valuable insights into lipid metabolism in specific neonatal populations, the lipidomic signatures of LPTB-LBW neonates—particularly their interplay with maternal lipid metabolism—remain largely unexplored, limiting our ability to identify early biomarkers and develop targeted therapeutic strategies for this high-risk subgroup.

To address this critical knowledge gap, we employed a broadly targeted lipidomics approach utilizing ultra-performance liquid chromatography-tandem mass spectrometry (UPLC-MS/MS) to comprehensively profile peripheral blood samples from neonates: LPTB-LBW cases and later preterm birth-normal birth weight (LPTB-NBW) controls. Our study aims to: (1) delineate the unique lipidomic signature of LPTB-LBW neonates, and (2) investigate transgenerational associations between maternal early-pregnancy lipids and neonatal lipid profiles.

## MATERIAL AND METHODS

### Study population and sample collection

This case-control study enrolled neonates delivered at a tertiary hospital in Beijing, China, between July 2019 and April 2023 (Fig. 1). The case group comprised LPTB-LBW neonates, defined as gestational age $34^0/^7$-$36^6/^7$ weeks with birth weight <2,500 g. The control group consisted of LPTB-NBW neonates matched by gestational age ($34^0/^7$-$36^6/^7$ weeks), sex, maternal age at delivery, and delivery hospital, with birth weights between 2,500–4,000 g. Exclusion criteria included maternal smoking or alcohol consumption during pregnancy, *in vitro* fertilization conception, multiple pregnancies, or congenital anomalies.

Clinico-demographic data were extracted from the hospital's electronic medical record system, including neonatal parameters: birth weight, delivery mode, and paternal age; Maternal characteristics: ethnicity, gravidity, parity, last menstrual period date, height, pre-pregnancy weight, pre-pregnancy body mass index (BMI), and weight at delivery; and

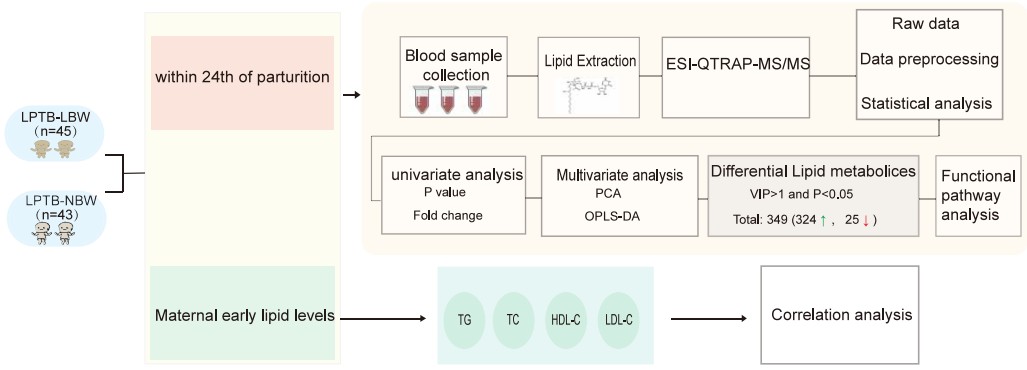

**Figure 1** Workflow of study.

perinatal factors: delivery season and maternal lipid profiles at 10–14 weeks gestation: total cholesterol (TC), triglycerides (TG), high density lipoprotein (HDLC), low density lipoprotein (LDLC).

Plasma samples were collected within 4 h postpartum prior to clinical interventions, aliquoted into sterile EP tubes and frozen at −80 °C until subsequent analysis.The study was approved by the Ethics Committee of China-Japan Friendship Hospital (number: 2023-KY-057). Parental informed consent was obtained in writing from each participant.

## Sample preparation

Plasma samples were thawed from −80 °C storage and vortexed for 10 s. A 50 µL aliquot of each sample was transferred into labeled centrifuge tubes. Lipid extraction was performed by adding one mL of lipid extraction solution (MTBE: MeOH = 3:1, V/V) containing the internal standards and vortexing the mixture for 15 min. After adding 200 µL of water, samples were vortexed for 1 min and centrifuged at 12,000 rpm for 10 min at 4 °C. The supernatant was collected and dried, followed by resuspension in 200 µL of lipid reconstitution solution (acetonitrile: isopropanol = 1:1, V/V) for LC-MS/MS analysis.

## UPLC-MS/MS

The lipidomic analysis was performed using an ExionLC AD Ultra Performance Liquid Chromatography (UPLC) system (SCIEX, https://sciex.com.cn/) coupled with a QTRAP®6500+ tandem mass spectrometer (SCIEX) as previously described (*Liu et al., 2025*). Chromatographic separation was conducted on a Thermo Accucore™ C30 column with a mobile phase consisting of acetonitrile/water (60/40, V/V) and acetonitrile/isopropanol (10/90, V/V), each containing 0.1% formic acid and ammonium formate. The flow rate was set to 0.35 mL/min, and the column was maintained at 45 °C. The gradient conditions are outlined in the detailed supplementary methods.

Mass spectrometry analysis employed electrospray ionization (ESI) s in positive and negative ionization modes. Optimized declustering potential (DP) and collision energy (CE) were applied for each lipid-specific multiple reaction monitoring (MRM) transition. Lipids were identified *via* retention time (RT) and precursor/product ion pairs matched

against the in-house MetWare database. Quantitation was performed in MRM mode, where precursor ions were isolated in Q1, fragmented in Q2, and characteristic product ions were filtered in Q3 to enhance specificity. Peak area integration of the chromatographic peaks for all detected ions was performed using MultiQuant software, with peak areas serving as the relative content of each lipid. A correction was applied to the peaks based on the lipid retention time and peak shape across different samples, ensuring accuracy in both qualitative and quantitative analysis. Missing values were addressed by filling them with one-fifth of the minimum value for each lipid. To ensure the reliability of the lipidomic data, quality control (QC) samples containing known concentrations of internal standards were analyzed (detailed information is provided in Table S1). The response variability of the internal standards, represented by the coefficient of variation (CV), was kept below 15%, indicating a stable and reliable analytical process. Total ion chromatograms (TICs) of QC samples were also used to assess the reproducibility of lipid extraction and detection. Figure S1 demonstrates the excellent repeatability of sample extraction and detection processes through the overlay of total ion chromatograms (TIC) from various QC samples.

## Statistical analysis

All statistical analyses were performed using R software (version 4.3.4; *R Core Team, 2023*). Baseline characteristics of neonates were summarized as means and standard deviations (mean ± SD) for continuous variables, and group comparisons were performed using the Mann–Whitney U test or Student's $t$-test. Categorical variables were presented as counts (percentages) and analyzed using the chi-square test. Unsupervised principal component analysis (PCA) was conducted using the prcomp function in R to explore the lipid metabolism profiles of the two groups. Differential lipid identification was based on both univariate and multivariate analyses. was utilized to execute Orthogonal partial least squares discriminant analysis (OPLS-DA) was performed using the MetaboAnalystR package in R to construct a predictive model and calculate Variable Importance in Projection (VIP) scores. Lipids with VIP >1, fold change (FC)>1.2 or <0.83 and $P$-value <0.05 were considered significant differential metabolites. Metabolic pathway enrichment analysis was performed using the Kyoto Encyclopedia of Genes and Genomes (KEGG) database. Spearman's correlation coefficient was used to evaluate associations between neonatal lipid levels and maternal lipid profiles during early pregnancy, including TC, TG, HDL-C, and LDL-C.

## RESULTS

### Clinical characteristics

A total of 88 neonates were included in the study, with 45 in the LPTB-LBW group and 43 in the LPTB-NBW group. Demographic characteristics are summarized in Table 1. No significant differences were observed between the two groups in gender, gestational age (249.27 ±6.58 days *vs.* 250.49 ±5.29 days, $p$=0.341) and parental variables, including maternal nationality, gravidity, parity, last menstrual period, height, weight at delivery, pre-pregnancy weight, pre-pregnancy BMI, BMI classification, delivery season, delivery mode, and paternal age (all $p$ > 0.05). The LPTB-LBW group had significantly lower birth

**Table 1  Clinical characteristics of the study subjects.**

| Variables | Overall (N = 88) | LPTB–LBW (n = 45) | LPTB–NBW (n = 43) | P |
|---|---|---|---|---|
| **Infant characteristics** | | | | |
| Gestational age (days), mean ± SD | 249.86 ± 5.98 | 249.27 ± 6.58 | 250.49 ± 5.29 | 0.341 |
| Birth weight (g), mean ± SD | 2,477.50 ± 371.10 | 2,194.11 ± 253.52 | 2,774.07 ± 204.54 | <0.001 |
| Sex = Male, n (%) | 50 (56.8) | 27 (60.0) | 23 (53.5) | 0.688 |
| Maternal characteristics | | | | |
| Nations = the Han plural, n (%) | 81 (92.0) | 41 (91.1) | 40 (93.0) | 1 |
| Age at delivery (years), mean ± SD | 32.70 ± 4.15 | 32.67 ± 4.52 | 32.74 ± 3.78 | 0.931 |
| Gravidity (times), mean ± SD | 1.95 ± 1.53 | 1.96 ± 1.80 | 1.95 ± 1.21 | 0.995 |
| Parity (times), mean ± SD | 1.32 ± 0.56 | 1.22 ± 0.42 | 1.42 ± 0.66 | 0.099 |
| Last menstruation season, n (%) | | | | 0.555 |
| Autumn | 21 (23.9) | 11 (24.4) | 10 (23.3) | |
| Spring | 15 (17.0) | 9 (20.0) | 6 (14.0) | |
| Summer | 31 (35.2) | 17 (37.8) | 14 (32.6) | |
| Winter | 21 (23.9) | 8 (17.8) | 13 (30.2) | |
| Hight (cm), mean ± SD | 163.00 ± 4.63 | 162.67 ± 4.38 | 163.35 ± 4.91 | 0.493 |
| Maternal weight when delivery (kg), mean ± SD | 68.69 ± 9.51 | 69.89 ± 9.78 | 67.43 ± 9.17 | 0.227 |
| Pre-pregnancy weight (kg), mean ± SD | 57.32 ± 7.68 | 58.10 ± 7.43 | 56.50 ± 7.95 | 0.331 |
| BMI (kg/m$^2$), mean ± SD | 21.96 ± 3.07 | 22.16 ± 2.55 | 21.74 ± 3.55 | 0.53 |
| BMI classification, n (%) | | | | 0.652 |
| Fat | 14 (15.9) | 7 (15.6) | 7 (16.3) | |
| Thin | 6 (6.8) | 2 (4.4) | 4 (9.3) | |
| Normal | 68 (77.3) | 36 (80.0) | 32 (74.4) | |
| Delivery season, n (%) | | | | 0.403 |
| Autumn | 25 (28.4) | 16 (35.6) | 9 (20.9) | |
| Spring | 20 (22.7) | 8 (17.8) | 12 (27.9) | |
| Summer | 26 (29.5) | 12 (26.7) | 14 (32.6) | |
| Winter | 17 (19.3) | 9 (20.0) | 8 (18.6) | |
| Delivery mode, n (%) | | | | 0.433 |
| Cesarean section | 47 (53.4) | 27 (60.0) | 20 (46.5) | |
| Obstetric forceps | 4 (4.5) | 2 (4.4) | 2 (4.7) | |
| Spontaneous delivery | 37 (42.0) | 16 (35.6) | 21 (48.8) | |
| Paternal age (years), mean ± SD | 34.02 ± 5.25 | 34.07 ± 5.43 | 33.98 ± 5.12 | 0.937 |

weight (2,194.11 ± 253.52 g *vs.* 2,774.07 ± 204.54 g, *p*<0.001) compared to the LPTB–NBW group.

## Widely targeted lipidomic profiling

Lipidomic profiling identified 1,173 lipid species classified into six major lipid classes: 510 glycerophospholipids (GP), 319 glycerolipids (GL), 235 sphingolipids (SP), 78 fatty acids (FA), 29 sterol lipids (ST), and two prenol lipids (PR) (Fig. 2).

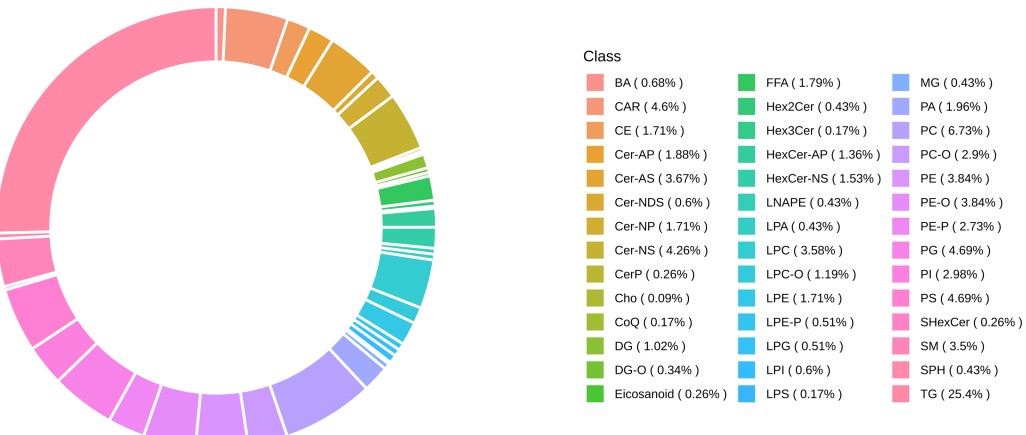

**Figure 2** Classification of identified 1,173 lipids.

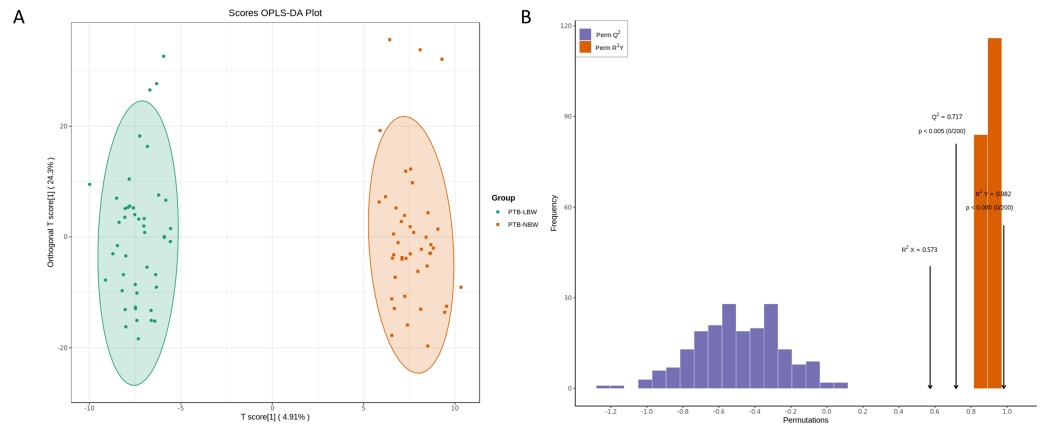

**Figure 3** Orthogonal partial least squares discriminant analysis (OPLS-DA) of the lipidomic data. (A) OPLS-DA score plots. (B) OPLS-DA validation plot.

## Differential lipid analysis

The PCA score plot revealed distinct clustering patterns between the two groups (Fig. S2). OPLS-DA analysis confirmed a significant separation (Fig. 3A), with model validation parameters indicating robust explanatory power and predictive reliability ($R^2X = 0.584$, $R^2Y = 0.978$, and $Q^2 = 0.651$; Fig. 3B).

A total of 349 lipids exhibited significant differential expression, including 324 up-regulated and 25 down-regulated lipids (Fig. 4A, Table S2). Among the 32 perturbed lipid subclasses, GL (162/349, 46.4%) were the most affected, followed by GP (87/349, 24.9%) and SP (75/349, 21.4%); (Fig. 4B).

As detailed in Fig. 4C and Table S2, comprehensive lipidomic profiling revealed distinct metabolic perturbations. All differentially expressed GLs (153/162, 94.4%) were upregulated, with long-chain polyunsaturated triglycerides (TG) constituting the predominant subclass. Striking upregulation was observed for TG (18:2_18:3_18:4) (FC

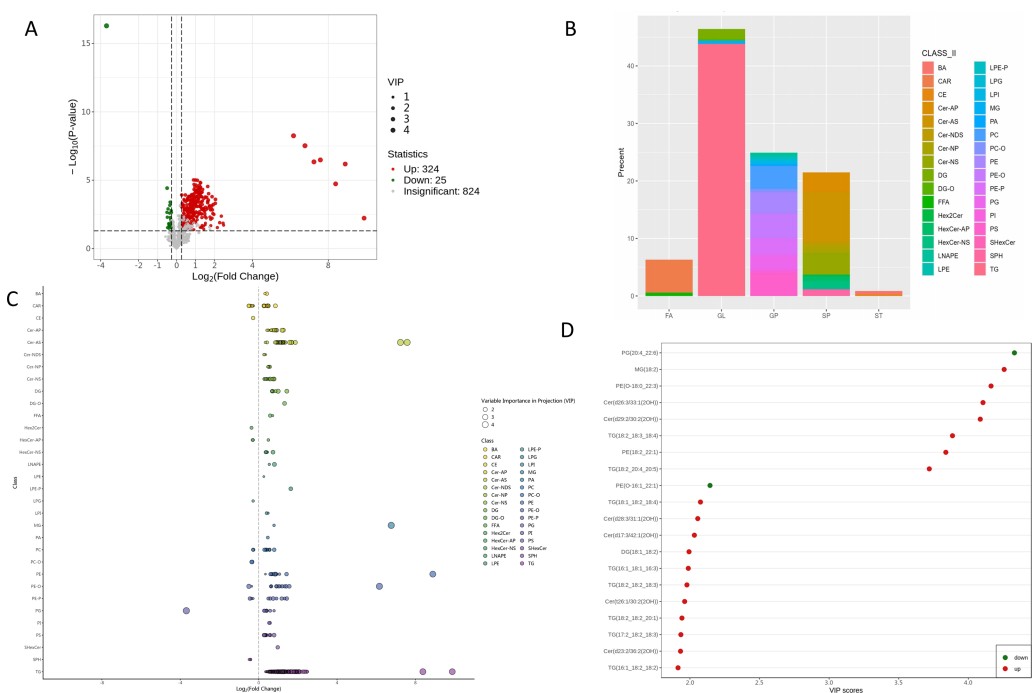

**Figure 4 Differential lipid profile.** (A) Volcano plot of identified lipids, showing log2 fold change (*x*-axis) and −log10 *P*-values (*y*-axis). Each point represents a lipid; green, red, and gray points indicate down-regulated, up-regulated, and non-significant lipids, respectively. Larger dots have higher VIP values. (B) Classification of identified 349 differential lipids. (C) Differential lipids scatter plot. (D) Top 20 differential lipids according to VIP score in the OPLS-DA model.

= 333.5, $p = 1.84 \times 10^{-5}$, VIP = 3.89) and TG (18:2_20:4_20:5) (FC = 943.6, $p = 0.0059$, VIP = 3.72). Monoglycerides (MG) and diglycerides (DG) were also significantly elevated, including MG (18:2) (FC = 108.9, $p = 3.00 \times 10^{-8}$, VIP = 4.26) and DG (18:1_18:2) (FC = 1.99, $p = 9.72 \times 10^{-6}$, VIP = 1.99).

Among sphingolipids, 82.7% (62/75) of differential ceramides showed marked upregulation. Key species included Cer (d26:3/33:1(2OH)) (FC = 191.14, $p = 3.28 \times 10^{-8}$, VIP = 4.11), Cer(d29:2/30:2(2OH)) (FC = 151.10, $p = 4.53 \times 10^{-7}$, VIP = 4.08), and Cer(d28:3/31:1(2OH)) (FC = 3.00, $p = 1.80 \times 10^{-4}$, VIP = 2.06). Conversely, sphingoid bases such as SPH(d18:1), SPH(d18:2), and SPH(d18:0) were uniformly downregulated.

The majority of GPs (76/87, 87.4%) exhibited upregulation. Phosphatidylethanolamines (PE) showed pronounced increases, exemplified by PE (18:2_22:1) (FC = 473.8, $p = 6.54 \times 10^{-7}$, VIP = 3.84) and PE (O-18:0_22:3) (FC=71.67, $p = 5.61 \times 10^{-9}$, VIP = 4.16). Conversely, phosphatidylglycerol PG (20:4_22:6) (FC = 0.08, $p = 5.13 \times 10^{-17}$, VIP = 4.33) and PE (O-16:1_22:1) (FC = 0.71, $p = 3.79 \times 10^{-5}$, VIP = 2.14) were significantly reduced.

Free fatty acids demonstrated significant upregulation in FFA(18:2) and FFA(20:5). Acylcarnitines, including C6:1 (FC = 1.81, $p = 0.003$, VIP = 1.81) and C8:1 (FC = 1.38, $p = 0.002$, VIP = 1.41), were predominantly elevated. Notably, two bile acids—glycocholic

**Table 2  Maternal serum lipid levels during early pregnancy.**

| Lipid levels | Overall (N = 88) | LPTB-LBW (n = 45) | LPTB-NBW (n = 43) | P |
|---|---|---|---|---|
| TC (mmol/L), mean ± SD | 4.46 ± 0.76 | 4.41 ± 0.68 | 4.51 ± 0.83 | 0.530 |
| TG (mmol/L), mean ± SD | 1.13 ± 0.38 | 1.10 ± 0.39 | 1.16 ± 0.38 | 0.476 |
| HDLC (mmol/L), mean ± SD | 1.56 ± 0.27 | 1.58 ± 0.22 | 1.54 ± 0.32 | 0.494 |
| LDLC (mmol/L), mean ± SD | 2.48 ± 0.45 | 2.46 ± 0.40 | 2.51 ± 0.51 | 0.630 |

Notes.

TC, total cholesterol; TG, triglycerides; HDLC, high density lipoprotein; LDLC, low density lipoprotein.

acid and taurocholic acid—sharked significant increases, while cholesteryl esters were consistently downregulated.

Among the top 20 VIP-ranked lipids, only two species—PG (20:4_22:6) and PE (O-16:1_22:1)—were downregulated, whereas others (*e.g.*, MG (18:2), PE (O-18:0_22:3), and Cer (d26:3/33:1(2OH))) showed robust upregulation (Fig. 4D). Notably, TG (18:2_20:4_20:5) demonstrated the most pronounced upregulation in the top 10 FC-ranked lipids (Fig. S3).

KEGG classification of differentially expressed lipids demonstrated that the metabolic pathways (ko01100) contained the highest proportion of dysregulated lipid species (92.7% of annotated lipids, Fig. 5A). Subsequent enrichment analysis further highlighted significant alterations in lipid-related metabolic processes, including: cholesterol metabolism (ko04979), glycerolipid metabolism (ko00561) and sphingolipid metabolism (ko00600) (Fig. 5B).

## Association between neonatal different lipid and maternal early lipid levels

As shown in Table 2, maternal lipid levels during early pregnancy—including TC, TG, HDL-C, and LDL-C—did not differ statistically between the two groups (all $p > 0.05$). Correlation analysis between maternal early lipids and neonatal top 50 VIP-ranked differential lipids revealed that HDLC was the only maternal lipid parameter significantly associated with neonatal lipid levels ($p < 0.05$) (Fig. 6). HDL-C exhibited negative correlations with specific neonatal lipids, particularly TG, DG, ceramides, and PE. The strongest inverse correlation was observed between maternal HDL-C and TG(16:0_18:2_18:2) ($r = -0.33$, $p = 0.002$) (Table S3). In contrast, maternal LDL-C, TC, and TG showed no significant correlations with neonatal lipids (all $p > 0.05$).

## DISCUSSION

This study delineates a unique lipidomic signature in LPTB-LBW neonates, characterized by glycerolipid overload, glycerophospholipid remodeling, sphingolipid imbalance, and sterol/fatty acid perturbations. These findings highlight a unique metabolic phenotype shaped by the dual insults of prematurity and IUGR, with maternal HDLC emerging as a novel modifier of fetal lipid programming.

The marked upregulation of GLs, particularly long-chain polyunsaturated triglycerides and monoglycerides, dominates the lipidomic profile of LPTB-LBW neonates. TGs such

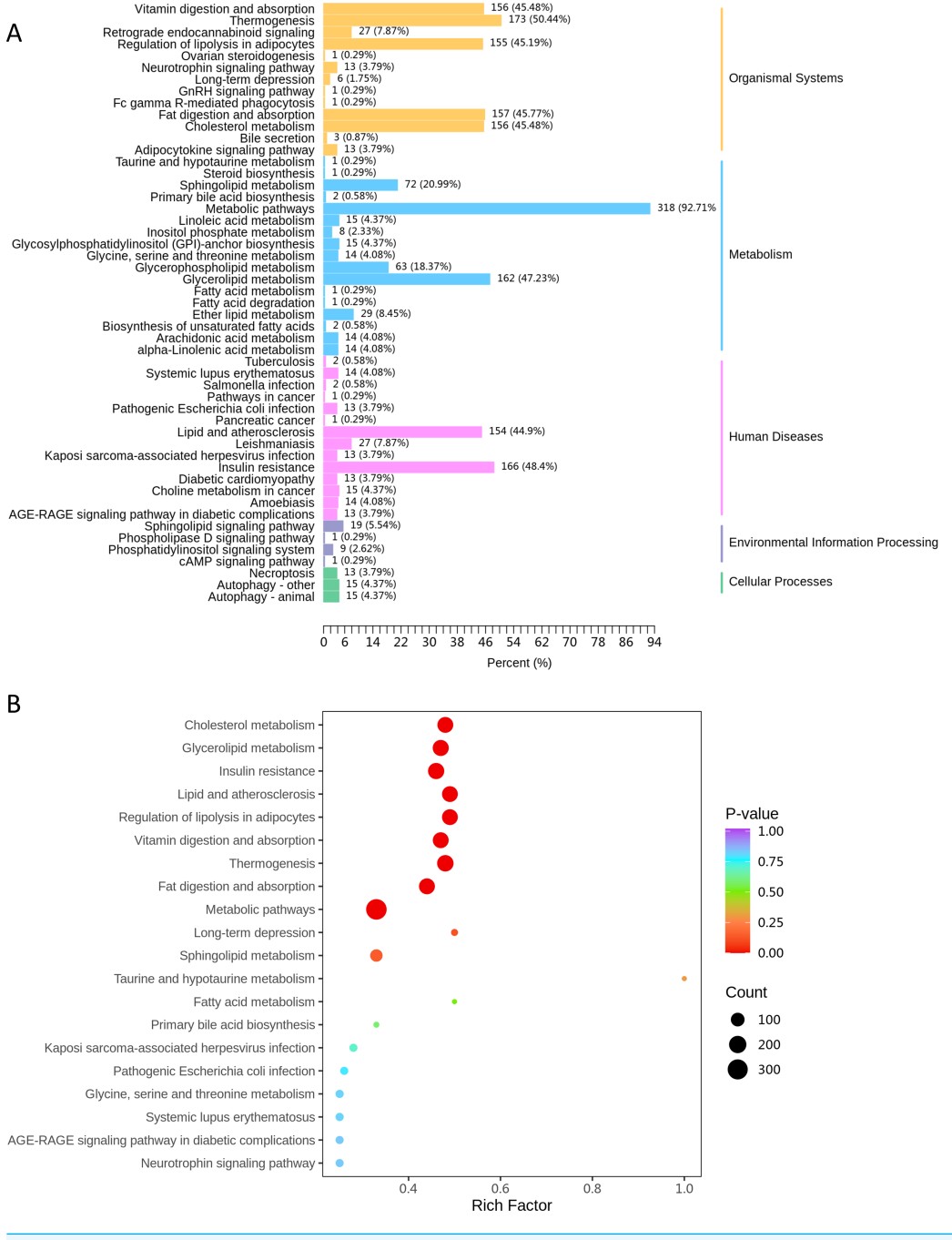

**Figure 5 The KEGG pathway analysis.** (A) KEGG classification chart, showing metabolic pathways annotated with differential lipids. The $y$-axis lists pathways, and the $x$-axis shows the number and proportion of annotated lipids. (B) KEGG enrichment bubble diagram, with pathway names ($y$-axis), Rich factor ($x$-axis), dot size (lipid count), and dot color ($P$-value significance, redder = more significant).

as TG (18:2_18:3_18:4) and TG (18:2_20:4_20:5) exhibited extreme elevations (FC > 300), suggesting a compensatory mechanism for energy storage under conditions of intrauterine nutrient deprivation. This aligns with IUGR-associated fetal adaptations,

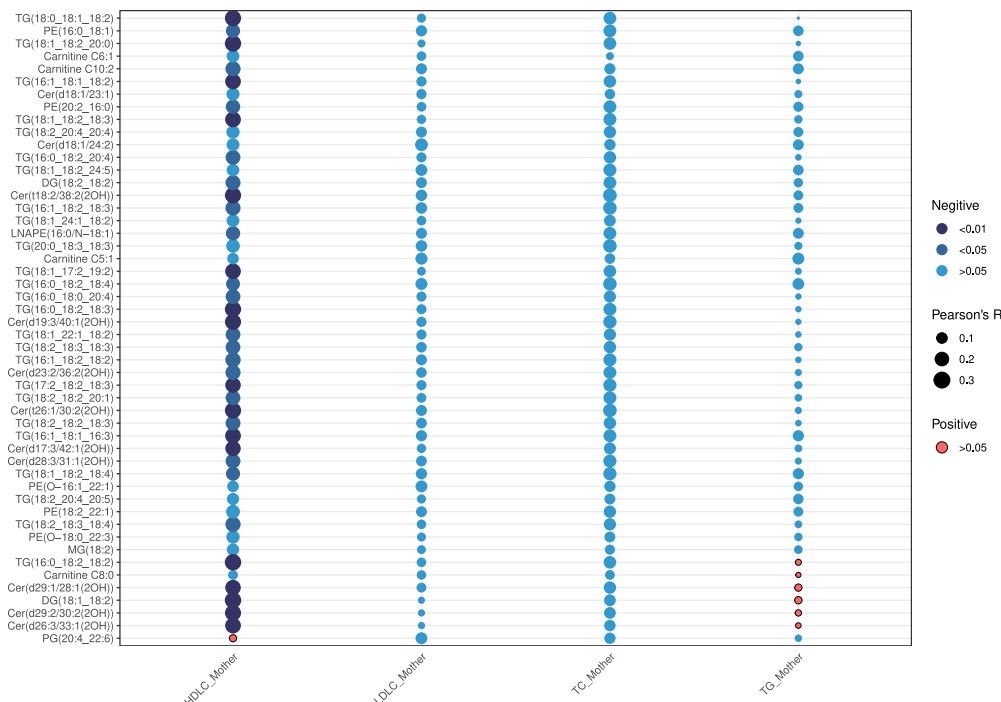

**Figure 6** Correlation analyses of the first 50 differential lipids and maternal lipid metabolism levels.

where limited glucose availability drives hepatic de novo lipogenesis and TG synthesis to preserve energy reserves (*Anam et al., 2022*; *Geidl-Flueck & Gerber, 2023*). However, excessive TG accumulation may reflect impaired mitochondrial $\beta$-oxidation, as evidenced by concurrent elevations in acylcarnitines (*e.g.*, C6:1, C8:1), indicative of incomplete fatty acid oxidation. The dominance of polyunsaturated TGs further implies altered desaturase activity, potentially modulated by placental insufficiency-induced hypoxia or maternal-fetal fatty acid transfer dysregulation. Notably, our findings contrast with term IUGR neonates who typically show TG depletion (*Miranda et al., 2018*). This dichotomy highlights gestational age-specific adaptations: where term IUGR mobilizes lipids for immediate energy needs, LPTB-LBW neonates prioritize lipid storage to buffer against extrauterine growth challenges. TG (18:2_20:4_20:5) demonstrated the highest fold-change (FC=943.6), positioning it as a candidate biomarker for metabolic surveillance.

The observed phosphatidylethanolamine elevation and phosphatidylglycerol depletion in LPTB-LBW neonates align with established mechanisms of membrane adaptation under developmental stress. As the second most abundant phospholipid in eukaryotic membranes, PE synergizes with phosphatidylcholine (PC) to maintain bilayer integrity and fluidity (*Dawaliby et al., 2016*; *Li et al., 2023*). Our findings demonstrate a significant upregulation of PE species (*e.g.*, PE (18:2_22:1)), which aligns with its essential function in maintaining membrane fluidity and supporting cellular proliferation under nutrient restriction. This compensatory elevation may facilitate membrane expansion required for

postnatal catch-up growth in LPTB-LBW neonates. In contrast, PG(20:4_22:6) exhibited a significant reduction in LPTB-LBW neonates, despite upregulation of other PG species. While PG constitutes only 1–2% of total phospholipids in most tissues, it is selectively enriched in pulmonary surfactant (accounting for up to 11% of alveolar hypophase lipids) and ranks second only to PC in maintaining alveolar stability (*Okano & Akino, 1979*; *Soll & Ozek, 2010*). PG(20:4_22:6) deficiency may impair surfactant function and mitochondrial integrity, though clinical sequelae (*e.g.*, respiratory dysfunction) require longitudinal validation.

The observed upregulation of ceramides and most glycosphingolipids, concomitant with a reduction in sphingosine, suggests a coordinated rewiring of sphingolipid metabolism in LPTB-LBW neonates. Ceramides serve as central intermediates in the biosynthesis and catabolism of all sphingolipids, including complex glycosphingolipids, acting as precursors for most sphingolipid species (*Summers, Chaurasia & Holland, 2019*). This metabolic shift may support cellular homeostasis and repair processes, potentially aiding in the maintenance of membrane integrity and nutrient recycling. However, this metabolic shift may facilitate cellular homeostasis and repair, excessive ceramide accumulation has been associated with the induction of apoptosis (*Chaurasia & Summers, 2021*; *Wilkerson et al., 2024*). Lipidomic screenings in large clinical cohorts have further revealed strong correlations between elevated serum and tissue levels of ceramides and/or dihydroceramides and obesity-related comorbidities, including insulin resistance, type 2 diabetes, and major adverse cardiovascular events (*Neeland et al., 2018*). Additionally, studies have identified a correlation between maternal obesity and altered ceramide cycling levels in both mothers and their offspring at 4 years of age (*León-Aguilar et al., 2019*). These findings emphasize the dual role of ceramides in neonatal development—supporting adaptive processes while posing potential pathological risks—and highlight the need for longitudinal studies to explore their long-term implications in LPTB-LBW populations.

In this study, we observed significant upregulation of two differential free fatty acids (FFAs), namely FFA(18:2) and FFA(20:5), alongside elevated levels of most acylcarnitines, such as C6:1 in LPTB-LBW neonates. The elevation of FFA(18:2) and FFA(20:5), both polyunsaturated fatty acids, may indicate heightened lipolysis or altered fatty acid mobilization to meet energy demands and support membrane biogenesis during catch-up growth. These FFAs serve as substrates for acylcarnitine synthesis, facilitating mitochondrial fatty acid $\beta$-oxidation (*Xiong, 2018*). The concurrent rise in medium-chain acylcarnitines (*e.g.*, C6:1 and C8:1) supports this hypothesis, as acylcarnitines shuttle fatty acids into mitochondria for energy production, a critical process in neonates facing nutritional deficits.

In parallel, the upregulation of GCA and TCA suggests enhanced bile acid synthesis, possibly driven by increased hepatic cholesterol catabolism *via* cytochrome P450 enzymes (*e.g.*, CYP7A1) (*van Best et al., 2020*). Bile acids play a pivotal role in lipid emulsification and absorption, potentially compensating for limited nutrient availability in LPTB-LBW neonates by optimizing intestinal efficiency. Conversely, the downregulation of cholesterol esters—a key storage form of cholesterol—may reflect a shift in cholesterol flux toward bile acid production rather than esterification. Cholesterol is an essential structural

 

component of cell membranes and a precursor for steroid hormones, transmembrane signaling molecules, and cellular proliferation (*Guo et al., 2024a*). Moreover, maternal dyslipidemia and elevated cholesterol biosynthesis during early pregnancy have been associated with an increased risk of delivering SGA neonates (*Kim et al., 2021*). This metabolic profile aligns with KEGG pathway annotations indicating active cholesterol metabolism, suggesting a compensatory mechanism to redirect cholesterol into bile acid synthesis under developmental constraints. The observed lipidomic shifts in LPTB-LBW neonates—marked by elevated phosphatidylethanolamine (PE), reduced PG(20:4_22:6), reprogrammed sphingolipid metabolism, and the differential regulation of free fatty acids (FFAs), acylcarnitines, bile acids, and cholesterol esters—collectively illustrate a systemic metabolic trade-off under developmental stress. The upregulation of polyunsaturated fatty acids (*e.g.*, FFA(18:2), FFA(20:5)) and acylcarnitines may prioritize energy metabolism and membrane fluidity to support catch-up growth, while increased bile acids and reduced cholesterol esters reflect hepatic adaptations to nutritional constraints. However, diminished cholesterol ester pools could impair steroidogenesis and membrane lipid reserves, potentially predisposing these neonates to long-term neuroendocrine and cardiovascular risks. These findings underscore the need for longitudinal studies to evaluate whether early lipidomic alterations increase susceptibility to metabolic comorbidities, such as insulin resistance or atherosclerosis, in later life.

Our study identifies a novel transgenerational axis linking maternal lipid metabolism to neonatal lipidomic reprogramming. Specifically, maternal first-trimester HDL-C levels exhibited a selective inverse correlation with neonatal atherogenic lipids—most prominently TG(16:0_18:2_18:2)—while maternal LDLC, TC, and TG showed no associations. During pregnancy, maternal metabolism undergoes significant adaptations to meet the nutritional and energy demands of both the mother and fetus. These metabolic adaptations include elevated levels of total cholesterol, low-density lipoprotein cholesterol (LDL-C), high-density lipoprotein cholesterol (HDL-C), and triglycerides, particularly during later stages of gestation (*Wiznitzer et al., 2009*). Placental trophoblasts play a crucial role in fetal development by absorbing glycerol and free fatty acids generated through enzyme-catalyzed hydrolysis of HDL and LDL. These lipids are subsequently re-esterified to form fats essential for fetal growth (*von Versen-Hoeynck & Powers, 2007*). HDL is critical for maintaining intra-follicular cholesterol homeostasis, a process essential for embryo development (*Fujimoto et al., 2009*). Studies have found that adjusted birth weight was negatively correlated with maternal HDLC levels starting from the 10th gestational week in overweight or obese women (*Misra, Trudeau & Perni, 2011*). Recent research has demonstrated an inverse relationship between maternal serum HDLC concentrations and birth weight at the 224th and 36th weeks of gestation. Elevated HDLC levels at the 36th gestational week have been associated with an increased risk of SGA neonates. Changes in maternal HDLC levels throughout pregnancy were linked to reduced neonatal size (*Wang et al., 2020*). Our findings reveal, for the first time, an association between maternal HDLC levels in early pregnancy and endogenous cholesterol biosynthesis, highlighting the link between lipid metabolism in LPTB-LBW neonates and maternal lipid profiles during early gestation.

Despite the use of real-case data and efforts to minimize biases, this study has certain limitations. These limitations include the single-center design and relatively small sample size, which may affect the generalizability of the findings. Additionally, the analysis of antenatal lipids was limited to a single time point in early pregnancy, restricting the ability to assess lipid changes throughout gestation. Future studies involving longitudinal sampling across preconception, pregnancy, and the postnatal period could provide deeper insights into lipid trajectory patterns and the causal relationships between LPTB-LBW lipids and maternal lipid levels.

## CONCLUSION

This study delineates the unique lipidomic signature of LPTB-LBW neonates, revealing synergistic disruptions in lipid metabolism driven by dual exposures to prematurity and intrauterine growth restriction. Key findings include hyperaccumulation of glycerolipids—notably long-chain polyunsaturated triglycerides—and ceramides, alongside altered glycerophospholipid profiles (elevated PE, reduced PG) and perturbed sphingolipid/cholesterol metabolism. Notably, maternal HDLC levels during early pregnancy inversely correlated with neonatal triglycerides and ceramides, implicating transgenerational lipid regulation as a modifiable risk factor. Our findings underscore the need for longitudinal studies to evaluate the clinical impact of these lipid alterations and explore targeted interventions to mitigate metabolic risks in this vulnerable population.

### Funding
This work was funded by National High Level Hospital Clinical Research Funding (2022-NHLHCRF-LX-01-0301), and the Clinical research project of Beijing research ward construction (2022-YJXBF-04-01-03). The funders had no role in study design, data collection and analysis, decision to publish, or preparation of the manuscript.

### Grant Disclosures
The following grant information was disclosed by the authors:
National High Level Hospital Clinical Research Funding: 2022-NHLHCRF-LX-01-0301.
The Clinical research project of Beijing research ward construction: 2022-YJXBF-04-01-03.

### Competing Interests
The authors declare there are no competing interests.

### Author Contributions
- Jing Liu conceived and designed the experiments, performed the experiments, prepared figures and/or tables, authored or reviewed drafts of the article, and approved the final draft.
- Lijuan Tang performed the experiments, prepared figures and/or tables, authored or reviewed drafts of the article, and approved the final draft.

- Qi Sun performed the experiments, analyzed the data, prepared figures and/or tables, authored or reviewed drafts of the article, and approved the final draft.
- Di Lv performed the experiments, analyzed the data, prepared figures and/or tables, authored or reviewed drafts of the article, and approved the final draft.
- Yuanmei Chen performed the experiments, analyzed the data, authored or reviewed drafts of the article, and approved the final draft.
- Fang Ye analyzed the data, authored or reviewed drafts of the article, and approved the final draft.
- Die Liu performed the experiments, analyzed the data, authored or reviewed drafts of the article, and approved the final draft.
- Qin Hui performed the experiments, analyzed the data, authored or reviewed drafts of the article, and approved the final draft.
- Haixiao Liang performed the experiments, analyzed the data, authored or reviewed drafts of the article, and approved the final draft.
- Chao Wang performed the experiments, analyzed the data, authored or reviewed drafts of the article, and approved the final draft.
- Qi Zhang conceived and designed the experiments, authored or reviewed drafts of the article, and approved the final draft.

## Human Ethics

The following information was supplied relating to ethical approvals (*i.e.*, approving body and any reference numbers):

The Ethics Committee of China-Japan Friendship Hospital

## Data Availability

The raw measurements are available in the Supplementary File.

## Supplemental Information

Supplemental information for this article can be found online at http://dx.doi.org/10.7717/peerj.19542#supplemental-information.

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
