# Peer review of "Dysregulated lipid metabolism in late preterm low birth weight neonates: A case-control study on maternal lipid levels and early metabolic programming implications"

_PeerJ, doi:10.7717/peerj.19542_

## Round 0.1 · original submission · Minor Revisions

Please revise following reviewer's advice.

Reviewer 1 ·

Basic reporting

This manuscript examines the differences in lipid profiles between late preterm birth-low birth weight (LPTB-LBW) and late preterm birth-normal birth weight (LPTB-NBW) neonates. Furthermore, it explores the correlations between the differential lipids and maternal lipid levels during early pregnancy. This manuscript has merit in analyzing plasma lipid levels between LPTB-LBW and LPTB-NBW infants, as well as in demonstrating their differential lipid profiles. However, I have several suggestions to improve this manuscript.
From a basic reporting perspective, figure captions should be written in detail. For example, the caption for Figure 2 states, "Classification of identified 1173 lipids," but it should clarify the sample groups from which these lipids were identified. Also, the caption for Figure 4 mentions "Differential lipid profile" but should clarify the sample groups between which the differential lipid profile is being compared.
The main figures should be presented in higher quality. The lipid class names in Figure 4C and the annotated lipid names in Figure 6 are difficult to read. I suggest providing higher-quality versions of the main figures throughout the manuscript.

Experimental design

I suggest the following comments regarding the experimental design. First, I could not find the detailed conditions for UPLC-MS/MS. The manuscript mentions that these conditions are outlined in the supplementary information, but I could not see them. Second, the authors should add a citation to line 115, as it is written as "as previously described" without any reference. Third, lipid identification and annotation are crucial for demonstrating differential lipid profiles between the two groups, given the diversity of lipids arising from the presence of isomers. I suggest clarifying and explaining how these lipids were identified and annotated.

Validity of the findings

The following suggestions need to be addressed to ensure the validity of the findings. First, in the LPTB-LBW group, numerous triacylglycerols, glycerophospholipids, and sphingolipids were significantly upregulated compared to the LPTB-NBW group. Notably, triacylglycerols comprise the majority of the differential lipids. The authors need to provide a biological perspective to explain why these lipid changes occurred. Second, PG(20:4_22:6) shows a significant decrease in the LPTB-LBW group compared to the LPTB-NBW group. The authors should clarify why this specific lipid shows a significant decrease in the LPTB-LBW group. Do polyunsaturated fatty acids, such as arachidonic acid (20:4) or docosahexaenoic acid (22:6), have any association with this decrease? The authors should justify why this specific lipid shows significant differences between the two groups.

Additional comments

1. I suggest using the term glycerolipid instead of glycolipid. Based on the lipid classification from Lipid Maps, TG and MG are in the glycerolipids (GL) category.
2. Line 53: “As According to” should be changed into “As according to”
3. Line 95: Hight-> height
4. Line 245: T These-> These
5. Line 251: various sphinglipses-> please clarify this word.
6. Figure 1: LBTP-LBW=> LPTB-LBW and LBTP-NBW-> LPTB-NBW

Reviewer 2 ·

Basic reporting

The authors are encouraged to complete Lipidomics Minimal Reporting Checklist. The correct names of the lipid species should be provided along with the final concentration.

Experimental design

Although the experimental design is ok, the correlation analysis provided in Fig 6 seems unnecessary, the authors are encouraged to conduct plasma lipidomics on maternal samples as well and then do the correlation between maternal and neonatal lipidomics profiles and birth outcomes.

Validity of the findings

Some of the findings need further validation, the lipid species that behaved quite differently in Fig4A and 4D. Similarly, the observation on the odd/atypical long chain base ceramides that were significantly different between the groups.

Additional comments

Detailed review comments are attached

Annotated reviews are not available for download in order to protect the identity of reviewers who chose to remain anonymous.

Reviewer 3 ·

Basic reporting

Feedback on the Manuscript
- I appreciate the scope of this work, as I have personally had a late-preterm low birth weight child born at 36+0 weeks gestation (Polyhydramnios during pregnancy). Thank you for your efforts to better understand this from a lipidomic perspective.
Strengths
1. Novelty and Relevance:
o The study addresses a critical gap in neonatal lipidomics, particularly for late preterm low birth weight infants.
o Findings could have clinical implications for early metabolic risk assessment.
2. Comprehensive Lipidomic Analysis:
o The study uses UPLC-MS/MS, a high-sensitivity technique, ensuring robust lipid identification.
o The identification of 349 significantly different lipids adds depth to neonatal lipidomics research.
3. Well-Structured Statistical Analysis:
o Use of multivariate analysis (OPLS-DA, PCA, VIP scores) strengthens lipidomic comparisons.
o KEGG pathway enrichment effectively links lipid changes to metabolic pathways.
4. Maternal Influence Consideration:
o The correlation between maternal lipid levels and neonatal lipid profiles adds a valuable perspective on fetal lipid metabolism.

Areas for Improvement
1. MRM transition transparency
o A list of MRM transitions or textual explanation of these (summarized by class) would be beneficial to understand the quality of the lipid assignment.
2. Figures and Data Interpretation:
o The text on figures is overall very small and hard to read. Edits would be much appreciated to increase readability (legends, labels, etc).
3. References:
o Provide a reference for: “Recent studies have reported that plasma ceramide (Cer) levels, particularly Cer 229 (d18:1/20:0), Cer (d18:1/22:0), Cer (d18:1/23:0), and Cer (d18:1/24:0), are significantly reduced 230 in overweight and obese mothers compared to those with normal body weight” line 228
4. More Context in the Introduction:
o While the introduction discusses LBW risks and lipid metabolism, it lacks a discussion on previous lipidomic studies in neonates.
o Adding references to existing neonatal lipidomic research would help contextualize the study’s novelty.
5. Language and Clarity:
o Some grammatical errors and awkward phrasing need revision for clarity and conciseness.
o Example: "The metabolic variations in identified differential lipids, such as glycolipids at birth, may relation with maternal early lipid levels during pregnancy."
Suggested revision: "The metabolic variations in identified differential lipids, such as glycolipids at birth, may be related to maternal early lipid levels during pregnancy."
o Formic acid (FA) and Fatty Acid (FA) use the same abbreviation in the text. Consider removing the abbreviation on formic acid.

Experimental design

Areas for improvement:
1. Triacylglycerol Structure Elucidation:
o How confident are the authors regarding the definitive acyl chain composition of TGs using MS/MS alone? Were any MS3 experiments conducted to validate these assignments? Some clarification regarding the dominant isomer population may be beneficial.
o Ex. The authors mention (among many others) TG(16:1_18:1_16:3) – using MS/MS and retention time alone, were these assignments validated with additional methods? What ion type was used for these experiments (i.e., Na+, NH4+, Li+ etc)? I am particularly interested in this assignment as FA 16:3 is a bit unusual (not impossible … just notable).
2. Fatty Acid Identification:
o What MS/MS parameters were used to identify FAs? Are assignments mased on observed m/z alone along with retention time matching? FAs notoriously do not fragment well under typical MS/MS conditions.
3. More Context on Internal Standard Normalization and Inclusion:
o Could the authors’ provide additional information regarding the specific internal standards used, their concentrations, and any normalizations used specific to these experiments?
o Including internal standards pre- lipid extraction is a great way to account for variations in extraction efficiency – additional clarification on their utilization in this work is needed and would be beneficial for readers.

Validity of the findings

Areas for improvement:
1. Sample Size Considerations:
o The sample size (n=88) is relatively small (though larger than most considering a difficult population to sample!), which may limit statistical power.
o A brief discussion on sample size limitations and potential future expansion could strengthen the manuscript.
2. Contextual Discussion:
o I understand this is not at a clinical stage, but could the authors comment on how early lipid screening could help prevent or even reverse metabolic risks in LPTB-LBW infants?
3. Placental Analysis:
o As a note to the authors – it would be interesting to assess lipidomic profiles across placentas as well with varying birth weights/outcomes

Additional comments

Summary of the Manuscript
Title: Widely Targeted Plasma Lipidomic Analysis of Late Preterm Low Birth Weight Infants
Objective
The study aims to assess whether late preterm birth-low birth weight (LPTB-LBW) neonates exhibit disrupted lipid metabolism compared to late preterm birth-normal birth weight (LPTB-NBW) neonates.
Methods
• Plasma lipidomic analysis was conducted using ultra-performance liquid chromatography-tandem mass spectrometry (UPLC-MS/MS).
• 88 plasma samples were analyzed (45 LPTB-LBW and 43 LPTB-NBW neonates).
• Univariate and multivariate statistical analyses were performed to identify differential lipid profiles.
• KEGG pathway enrichment analysis was used to explore metabolic pathway perturbations.
• Correlation analysis assessed the relationship between neonatal lipid levels and maternal lipid levels during early pregnancy.
Key Findings
• 1173 lipids were identified, categorized into six major lipid classes.
• 349 significantly different lipids were identified, with 324 upregulated and 25 downregulated in the LPTB-LBW group.
• Increased lipid classes in LPTB-LBW:
o Glycolipids (e.g., TG(18:2_18:3_18:4), TG(18:2_20:4_20:5))
o Glycerophospholipids (e.g., PE(O-18:0_22:3), PE(18:2_22:1))
o Sphingolipids (e.g., Cer(d26:3/33:1(2OH)), Cer(d28:3/31:1(2OH)))
• Decreased lipid class in LPTB-LBW:
o Glycerophospholipids (e.g., PG (20:4_22:6))
• KEGG pathway analysis showed enrichment in cholesterol metabolism, glycerolipid metabolism, and regulation of lipolysis in adipocytes.
• TG (16:1_18:1_16:3) was negatively correlated with maternal early lipid levels, indicating a possible maternal influence on neonatal lipid metabolism.
Conclusions
• The study demonstrates distinct plasma lipidomic profiles between LPTB-LBW and LPTB-NBW neonates.
• Glycolipid and sphingolipid metabolism alterations may play a crucial role in metabolic disruptions associated with LPTB-LBW infants.
• Findings highlight the importance of maternal lipid metabolism in influencing neonatal lipid profiles.
* * *
Strengths
1. Novelty and Relevance:
o The study addresses a critical gap in neonatal lipidomics, particularly for late preterm low birth weight infants.
o Findings could have clinical implications for early metabolic risk assessment.
2. Comprehensive Lipidomic Analysis:
o The study uses UPLC-MS/MS, a high-sensitivity technique, ensuring robust lipid identification.
o The identification of 349 significantly different lipids adds depth to neonatal lipidomics research.
3. Well-Structured Statistical Analysis:
o Use of multivariate analysis (OPLS-DA, PCA, VIP scores) strengthens lipidomic comparisons.
o KEGG pathway enrichment effectively links lipid changes to metabolic pathways.
4. Maternal Influence Consideration:
o The correlation between maternal lipid levels and neonatal lipid profiles adds a valuable perspective on fetal lipid metabolism.

---

## Round 0.2 · accepted · Accept

Congratulations. it is well revised.

Reviewer 1 ·

Basic reporting

no comment

Experimental design

no comment

Validity of the findings

no comment

Additional comments

The authors have addressed my comments.

Reviewer 2 ·

Basic reporting

All concerns have been addressed.

Experimental design

Adressed.

Validity of the findings

NA

Additional comments

NA